# Risk Role of Breast Cancer in Association with Human Papilloma Virus among Female Population in Taiwan: A Nationwide Population-Based Cohort Study

**DOI:** 10.3390/healthcare10112235

**Published:** 2022-11-08

**Authors:** Chia-Hsin Liu, Chi-You Liao, Ming-Hsin Yeh, James Cheng-Chung Wei

**Affiliations:** 1Department of Surgery, Chung Shan Medical University Hospital, Taichung 40201, Taiwan; 2Division of Breast Surgery, Department of Surgery, Chung Shan Medical University Hospital, Taichung 40201, Taiwan; 3Institute of Medicine, College of Medicine, Chung Shan Medical University, Taichung 40201, Taiwan; 4Department of Allergy, Immunology & Rheumatology, Chung Shan Medical University Hospital, Taichung 40201, Taiwan; 5Graduate Institute of Integrated Medicine, China Medical University Hospital, Taichung 40459, Taiwan

**Keywords:** human papilloma virus, breast cancer, Asian women

## Abstract

Purpose: We analyzed data from the National Health Insurance Research Database (NHIRD) in Taiwan, collected information regarding human papillomavirus (HPV) and breast cancer prevalence, and explored the association between HPV infection and the risk of breast carcinoma. Methods: We included the NHIRD data of 30,936 insured patients aged 20 years an older without breast cancer prior to the index date (date of HPV diagnosis) and matched each patient with a reference subject according to age, comorbidities, and index year (1:1 ratio). We calculated the incidence rates of breast cancer in the cohorts, age groups, and comorbidity groups, as well as the relative risk of breast cancer stratified by age and comorbidity in the HPV and non-HPV groups. Results: The patients with and without HPV had incidence rates of 12.5 and 9.81 per 10,000 person years, respectively. The risk of breast cancer for the 50−64 and ≥65 age groups was 1.67 and 1.36 times higher than that in patients younger than 49 years, respectively, and hypertension, chronic obstructive pulmonary disease, and diabetes mellitus were significant risk factors for breast cancer. The HPV group had a higher risk of developing breast cancer than the non-HPV group, regardless of age group and the presence or absence of comorbidities. Patients with HPV in the 50–64 age group were 1.39 times more likely to develop breast cancer than patients of the same age without HPV. Conclusion: Patients older than 49 were more likely to develop breast cancer, and patients with HPV had a higher likelihood of developing breast cancer, regardless of age and the presence or absence of comorbidities. HPV likely plays a causal role in breast cancer.

## 1. Introduction

Breast cancer is one of the cancers with the highest global prevalence among women and is the leading cause of death from cancer in women. According to American Cancer Society, breast cancer is the most common cancer in women in the United States, accounting for about 30% (or 1 in 3) of all new female cancers each year. In Taiwan, the incidence of invasive breast cancer increased from 66.1/100,000 in 2007 to 107.2/100,000 in 2018 [1]. However, the etiology of breast cancer remains unclear, generally referred to as a multifactorial disease. The established risk factors for breast cancer include age, obesity, early menarche, late menopause, alcohol consumption, no breast feeding, estrogen use, and a BRCA genetic mutation (BRCA1 and BRCA2, which produce proteins that help repair damaged DNA; people who inherit harmful variants in one of these genes have increased risks of several cancers, especially breast cancer and ovarian cancer.) [2,3]. Most of these factors are related to prolonged estrogen and progesterone stimulation. Other potential risk factors for breast cancer, such as insomnia, air pollution, and viral infection, have not been verified, and the results of studies on these risk factors remain contentious. Only 5–10% of patients with breast cancer have a family history of germline mutation, and the other 90–95% of cases are sporadic [2]. Therefore, determining potential risk factors for breast cancer is essential.

Human papillomavirus (HPV) infection is a sexually transmitted disease with the highest prevalence among both women and men. The CDC (Center for Disease Control and Prevention) reported that HPV is the most common STI, with about 43 million HPV infections in 2018, many among people in their late teens and early 20s. There are many different types of HPV. Some types can cause health problems, including genital warts and cancers. Approximately 90% of HPV infections cause no evident symptoms and resolve within 2 years [4]. However, some HPV infections persist and cause warts or sustained inflammation lesions [5]. HPV has also been established as a malignancy precursor or a risk factor for many cancers, including head and neck squamous cell carcinoma and cervical, anal, penile, bladder, laryngeal, and esophageal cancers [6,7]. In the HPV family, high-risk HPV types, including HPV16 and HPV18, have particularly been associated with an increased risk of cancer. The E6 and E7 proteins in HPV promote cellular proliferation and inactivate cell cycle checkpoint function, promoting viral proliferation by degrading p53 and Rb tumor suppressors. The E7 protein causes replication stress and oncogene-induced senescence. Oncogenesis can be directly induced by the virus protein [7].

In the United States, where the epidemiology of HPV infection was first established, the HPV prevalence is much higher among women than among men [8]. Furthermore, HPV has a notable carcinogenic effect on women’s fertility systems. Therefore, numerous studies have investigated the association between HPV infection and breast cancer. Most of these studies have sought to do so by detecting HPV DNA in both cancer and benign lesions in breast tissue through initial polymerase chain reaction (PCR) amplification by using standard, commercially available primers commonly used to screen HPV in cervical tissue or applied primers for specific oncogenic types. Several reports have indicated an association between HPV and breast cancer lesions [9,10,11,12,13,14,15,16,17,18,19,20,21,22,23], whereas others have disagreed [24,25,26,27,28,29,30,31,32]. Retrospective studies generally compare carcinoma with benign lesions. In these retrospective studies, several types of HPV (6, 11, 16, 18, and 33) have been detected in up to 86.21% of breast cancer tissue. However, most studies have collected data from patients with benign breast lesions, such as fibroadenoma, and have used fibrocystic change as a control group. Therefore, the conclusions of these studies are not robust.

The aim of our study was to analyze data from the National Health Insurance Research Database (NHIRD) in Taiwan, collect information regarding HPV and breast cancer prevalence, and explore the association between HPV infection and the risk of breast carcinoma.

## 2. Methods

### 2.1. Data Source and Study Population

Taiwan launched a single-payer National Health Insurance program in 1995. Since that time, 99% of the residents of Taiwan have been enrolled in the program. The NHIRD was subsequently established. The Longitudinal Health Insurance Database (LHID) contains a subset of the NHIRD data. One million patients insured through the National Health Insurance program were randomly selected to become part of the LHID, and their registration and original claims data were recorded. The diseases recorded in the database were defined according to the International Classification of Diseases, Ninth Revision, Clinical Modification (ICD-9-CM). We conducted a cohort study using data from the LHID from 2000 to 2013. We included insured patients aged 20 years and older without breast cancer prior to the index date, which was the date of HPV diagnosis (ICD-9-CM codes 079.4, 078.1, 795.05, 795.09, 795.15, 795.19, 795.75, and 796.79). A total of 61,872 patients were included and divided into HPV and non-HPV groups. The patients with HPV were matched to a reference subject according to age, comorbidities, and index year (1:1 ratio).

### 2.2. Main Outcome and Covariates

Breast cancer (ICD-9-CM code 174) was the study end point. The follow-up time was the interval between the index date and the date of breast cancer diagnosis, missing data, death, or the end of 2013. We also considered comorbidities related to breast cancer, including hypertension (ICD-9-CM codes 401–405, A260, and A269), coronary artery disease (ICD-9-CM codes 410–414, A270, and A279), chronic obstructive pulmonary disease (ICD-9-CM codes 491, 492, and 496), stroke (ICD-9-CM codes 43 and A29), diabetes mellitus (ICD-9-CM codes 250 and A181), hyperlipidemia (ICD-9-CM codes 272 and A182), and obesity (ICD-9-CM codes 278 and A183).

### 2.3. Statistical Analysis

The chi-square test was used to compare the proportional distribution of the age groups (20–49, 50–64, and ≥65) and comorbidities in the HPV and non-HPV cohorts. The mean ages of the groups were determined using a two-sample *t* test. We calculated the incidence rates of breast cancer in the cohorts, age groups, and comorbidity groups. The hazard ratio of each variable was estimated using a Cox regression model. We adjusted the hazard ratios by including the variables that were significant in the univariate Cox regression model in a multivariate Cox model. The relative risk of breast cancer, stratified by age and comorbidity in the HPV compared with the non-HPV groups, was also analyzed. The Kaplan–Meier method was used to assess the cumulative incidence of breast cancer in the cohorts, and the differences were compared using a log-rank test. Significance was set at a *p* value of <0.05.

## 3. Results

The distributions of the baseline characteristics in the case and control groups are presented in Table 1. The average follow-up time for the case cohort was 7.22 years (standard deviation (SD) = 3.67), and that for the control cohort was 7.12 years (SD = 3.69). The age and comorbidity distributions between the groups were similar. The mean age for both groups was 41.5 years. Hypertension and hyperlipidemia were the most common comorbidities in the groups. The incidence rates and relative risks of breast cancer in the study population are listed in Table 2. The patients with HPV had an incidence rate of 12.5 per 10,000 person years, and those without HPV had an incidence rate of 9.81 per 10,000 person years. The relative risk of breast cancer in the HPV group compared with the non-HPV group was 1.27 (95% confidence interval (CI) = 1.20–1.34). People older than 49 years were more likely to develop breast cancer. The risk of breast cancer in the 50−64 and ≥65 age groups was 1.67 times (95% CI = 1.56–1.80) and 1.36 times (95% CI = 1.32–1.52) times higher than that in patients younger than 49 years, respectively. Hypertension, chronic obstructive pulmonary disease, and diabetes mellitus were significant risk factors for breast cancer. The adjusted hazard ratios for hypertension, chronic obstructive pulmonary disease, and diabetes mellitus were 1.30 (95% CI = 1.20–1.41), 1.12 (95% CI = 1.01–1.24), and 1.90 (95% CI = 1.69–2.14), respectively. The results regarding the risk of breast cancer in the case and control groups are presented in Table 3. The HPV group had a higher risk of developing breast cancer than the non-HPV group, regardless of age group and the presence or absence of comorbidities. Moreover, the patients with HPV in the 50–64 age group had a 1.39 times higher risk of breast cancer than patients without HPV in the same age group, which represented the highest risk ratio between the groups. The results of the Kaplan–Meier analysis of breast cancer between the groups are presented in Figure 1. The patients with HPV had a significantly higher cumulative incidence of breast cancer than patients without HPV according to the log-rank test, with a *p* value of <0.05.

## 4. Discussion

The global incidence and mortality rates of breast cancer are increasing. Because the etiology of breast cancer is complex, identifying new potential risk factors for breast cancer is essential for both its treatment and prevention. HPV is transmitted through sustained skin-to-skin contact. The virus infects the epithelium and mucosa, inducing inflammation and production of oncogenic protein. This process of transmission and its carcinogenicity renders HPV a highly suspected risk factor for breast cancer development. Viral infection causes 18–20% of cancers [33], and the high-risk HPV16 and 18 are present in 50–70% of cervical cancer cases [34] and have been reported to be linked to anal, penile, and head and neck cancers [33]. Furthermore, HPV is present in 35.7% of lung cancer cases [35,36].

In this study, we paired the data of 30,936 women reported to be infected with HPV between 2000 and 2013 in the NHIRD with those of a non-HPV group, with an average follow-up of 7.2 years. We demonstrated that the hazard ratio (HR) for HPV in women aged between 20 and 49 was 1.27 (95% CI = 1.20–1.34), with a relative risk of 27%. Among women aged 50–64, the adjusted ratio was 1.39 (1.24, 1.57), with a 39% increased relative risk; among women aged 20–49, the adjusted ratio was 1.21 (1.14, 1.29), with a 21% increased relative risk. The adjusted ratio was 1.36 (1.15, 1.61), with a 36% increased relative risk, among women aged >65 years. For all ages, women who were infected with HPV had a 21%–36% increased risk of breast cancer compared with women without HPV. The incidence rate of breast cancer in women with HPV was 12.5/10,000 person years, and that in women without HPV was 9.81/10,000 person years. The breast cancer incidence for all women in Taiwan from 2010 to 2015 ranged between 6.32/10,000 and 7.3/10,000 after age standardization.

Most studies on HPV and breast cancer test HPV DNA in both benign breast lesions and breast cancer lesions [9,10,11,12,13,14,15,16,17,18,19,20,21,22,23,24,25,26,27,28,29,30,31,32]. Mohammad et al. conducted a meta-analysis in Iran including 11 studies with a total of 858 samples. The pooled odds ratio of developing breast cancer between women with and without HPV was 5.7 (0.7–46.8), and the total prevalence of HPV infection among women with breast cancer was 23.6%, which increased to 29.2% after exclusion of a heterogenous study.

Saimul et al. conducted a retrospective study in India in which they analyzed benign and malignant breast tissue. The results revealed a high frequency of HPV infection (63.9% in pretherapeutic women and 71% in women who received neoadjuvant therapy) in breast cancer tissue and only a 9.5% frequency in adjacent normal breast tissue. HPV16 and HPV18 were responsible for 69% and 35% of the infections, respectively, and HPV33 was responsible for 2.9% of the infections. In the same study, of the women with HIV, 30% had fibroadenomas, and 71.5% had benign phylloides, whereas only 9.5% of the normal women were HIV-infected.

In a 2017 review of the molecular evidence of HPV and breast cancer, Carolina et al. included 43 studies that tested HPV in breast cancer tissue with and without normal breast tissue. Most of the studies employed initial PCR amplification using commercially available primers to screen HPV in cervical cancer. Some studies further obtained primers from the E6 and E7 regions or primers sequenced to identify specific subtypes. The percentage of women who had breast cancer who were also identified as infected with HPV ranged from 0% to 86.2%, and the percentage ranged from 0% to 32% in women with nonmalignant tumors. The included studies analyzed both benign and malignant specimens, with 17 studies achieving statistical significance, compared to 6 studies in which statistical significance was not achieved. The heterogeneity maybe due to racial differences, age differences, a lack of standardization in optimizing DNA amplification, or differing epidemiologies between counties. Because the prevalence of HPV among women aged older than 18 was 7.3% in the United States from 2011 to 2014, 4.1% to 17.2% in different parts of Asia, and 2.1%–6.1% in different parts of Europe, the results of these studies must be interpreted with caution.

Our study is the first to apply data from a national health program to explore the effects of HPV on breast cancer by matching the data of women with HPV with data of women without HPV of the same age and with the same comorbidities. Atique et al. also used the NHIRD to explore the risk of breast cancer in women with HPV [13]. However, their study did not pair samples with similar comorbidities for comparison.

Most studies that have compared breast cancer and normal breast specimens have investigated HPV genotypes 16, 18, and 33. Three types of HPV vaccines are commercially available. Cervarix protects against initial infection with HPV 16 and 18; Gardasil protects against HPV 6, 11, 16, and 18; and protects against HPV 6, 11, 16, 18, 31, 33, 45, 52, and 58. However, these vaccines primarily benefit women who have not been exposed to HPV. The actual effectiveness and protective effects against breast cancer of HPV vaccines remain unclear and warrant further investigation.

Our result of a 27% total increased relative risk among women with HPV infection supports that the risk of breast cancer increases with HPV infection.

### 4.1. Limitations

The limitations of this study include its observational nature, selection biases related to the data being obtained from a registry of patients screened for suspected diseases rather than an unselected population-based cohort, and the lack of information on behavioral risk factors reported to affect risk of infection. Furthermore, we included no subgroups for other risk factors for breast cancer, such as insomnia, air pollution exposure, and other viral infections.

Because 90% of HPV infections have no observable symptoms and resolve within 2 years, the number of infected individuals may have been underestimated. The average follow-up was 7.22 years. New cases of infection may have occurred after that period.

### 4.2. Future Work

The prevalence of HPV in breast cancer indicates that prophylactic vaccination against HPV is required to limit infection in women. Moreover, the presence of HPV in breast tissue could be a powerful biomarker that could be implemented in a treatment protocol for HPV-infected breast cancer. Furthermore, HPV vaccines primarily benefit women who have not been exposed to HPV. Therefore, the vaccine’s actual effectiveness and protective effects against breast cancer remain unclear and warrant further investigation.

## 5. Conclusions

The strengths of our study include a large sample size and robust event rate, analytic control for known risk factors in the matched cohort, and independent confirmation of the magnitude and direction in the unmatched cohort.

Several HPV families, such as HPV16 and HPV18, have been linked to an increased risk of several types of cancer. In our study, we discovered that these families increase the risk of breast cancer.

People who were older than 49 years had a greater risk of developing breast cancer. The risk of breast cancer for the 50−64 and the >65 age groups was 1.67 and 1.36 times higher than that of people younger than 49 years, respectively. The patients with HPV had a higher risk of breast cancer than patients without HPV, regardless of age group and the presence or absence of comorbidities. Moreover, the patients with HPV in the 50–64 age group were 1.39 times more likely to develop breast cancer than the patients without HPV in the same age group. HPV may play a causal role in breast cancer.

According to the CDC, HPV is the most common sexually transmitted infection. When used correctly and consistently, condoms represent one of the most effective methods of protection. HPV vaccines primarily benefit women who have not been exposed to HPV. Therefore, the vaccine’s actual effectiveness and protective effects against breast cancer remain unclear and warrant further investigation.

## Figures and Tables

**Figure 1 healthcare-10-02235-f001:**
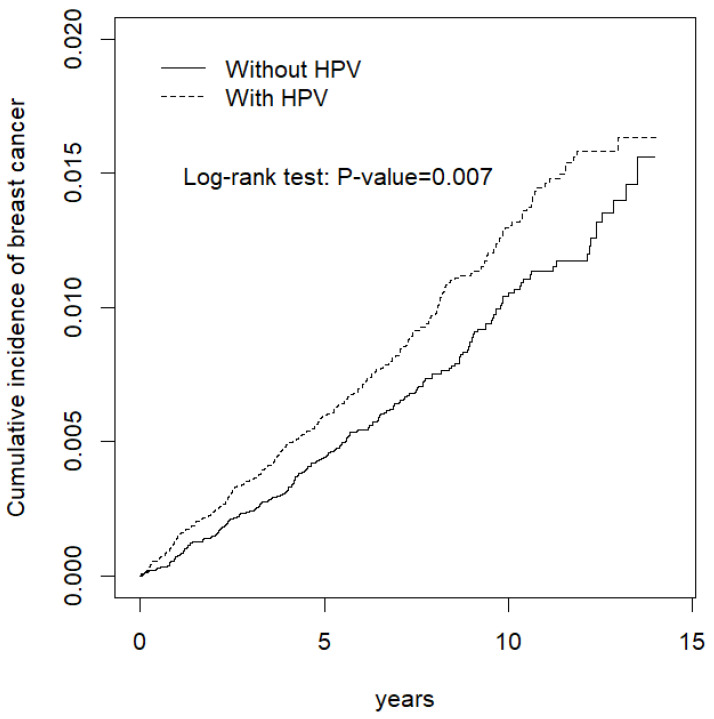
Cumulative incidence of breast cancer in individuals with and without HPV.

**Table 1 healthcare-10-02235-t001:** Baseline characteristics of individuals with and without HPV.

	HPV				
	No		Yes		
	*n* = 30,936		*n* = 30,936		
	*n*	%	*n*	%	*p*-Value
Age, year					0.99
20−49	22,560	72.9	22,560	72.9	
50−64	5583	18.1	5583	18.1	
≥65	2793	9.03	2793	9.03	
Mean (SD) ^†^	41.5	15.3	41.5	15.4	0.76
Comorbidity					
Hypertension	5102	16.5	5102	16.5	0.99
Coronary artery disease (CAD)	2681	8.67	2681	8.67	0.99
Chronic obstructive pulmonary disease (COPD)	1798	5.81	1798	5.81	0.99
Stroke	259	0.84	259	0.84	0.99
Diabetes mellitus	808	2.61	808	2.61	0.99
Hyperlipidemia	4697	15.2	4697	15.2	0.99
Obesity	560	1.81	560	1.81	0.99

Chi-square test; ^†^ two-sample *t* test; case group follow-up time: 7.22 (SD = 3.67); control group follow-up time: 7.12 (SD = 3.69).

**Table 2 healthcare-10-02235-t002:** Incidence of and risk factors for breast cancer.

	Event	PY	Rate ^#^	Crude HR (95% CI)	Adjusted HR ^&^ (95% CI)
HPV					
No	217	221,132	9.81	1.00	1.00
Yes	280	223,329	12.5	1.28(1.21, 1.35) **	1.27(1.20, 1.34) **
Age, years					
20−49	302	334,656	9.02	1.00	1.00
50−64	134	75,192	17.8	1.97(1.86, 2.10) **	1.67(1.56, 1.80) **
≥65	61	34,612	17.6	1.95(1.80, 2.12) **	1.36(1.23, 1.52) **
Comorbidity					
Hypertension					
No	368	376,435	9.78	1.00	1.00
Yes	129	68,026	19.0	1.94(1.83, 2.06) **	1.30(1.20, 1.41) **
Coronary artery disease (CAD)					
No	430	408,266	10.5	1.00	1.00
Yes	67	36,195	18.5	1.76(1.63, 1.90) **	1.02(0.93, 1.12)
Chronic obstructive pulmonary disease (COPD)					
No	457	421,435	10.8	1.00	1.00
Yes	40	23,026	17.4	1.60(1.45, 1.76) **	1.12(1.01, 1.24) *
Stroke					
No	491	441,751	11.1	1.00	1.00
Yes	6	2710	22.1	1.99(1.57, 2.54) **	1.04(0.82, 1.34)
Diabetes mellitus					
No	465	434,513	10.7	1.00	1.00
Yes	32	9948	32.2	3.01(2.70, 3.35) **	1.90(1.69, 2.14) **
Hyperlipidemia					
No	390	381,171	10.2	1.00	1.00
Yes	107	62,690	17.1	1.67(1.57, 1.78) **	1.03(0.95, 1.11)
Obesity					
No	488	437,968	11.1	1.00	1.00
Yes	9	6493	13.9	1.24(1.02, 1.52) *	1.14(0.94, 1.39)

CI, confidence interval; HR, hazard ratio; PY, person years; Rate ^#^, incidence rate per 10,000 person years; ^&^ multivariable analyses included age and comorbidities of hypertension CAD, COPD, stroke, diabetes mellitus, hyperlipidemia, and obesity; * *p* < 0.05, ** *p* < 0.001.

**Table 3 healthcare-10-02235-t003:** Incidence and HRs of breast cancer in individuals with and without HPV.

	HPV							
	No			Yes				
	Event	PY	Rate ^#^	Event	PY	Rate ^#^	Crude HR (95% CI)	Adjusted HR ^&^ (95% CI)
Age, years								
20−49	136	166,722	8.16	166	167,934	9.88	1.21(1.14, 1.29) **	1.21(1.14, 1.29) **
50−64	56	37,616	14.9	78	37,576	20.8	1.39(1.24, 1.57) **	1.39(1.24, 1.57) **
≥65	25	16,793	14.9	36	17,819	20.2	1.36(1.14, 1.61) **	1.36(1.15, 1.61) **
Comorbidity ^§^								
No	129	164,589	7.84	179	165,767	10.8	1.38(1.29, 1.47) **	1.38(1.29, 1.47) **
Yes	88	56,543	15.6	101	57,562	17.6	1.13(1.03, 1.24) *	1.12(1.02, 1.24) *

CI, confidence interval; HR, hazard ratio; PY, person years; Rate ^#^, incidence rate per 10,000 person years; ^&^ multivariable analyses included age and comorbidities of hypertension, CAD, COPD, stroke, diabetes mellitus, hyperlipidemia, and obesity; ^§^ individuals with hypertension, CAD, COPD, stroke, diabetes mellitus, hyperlipidemia, and obesity were classified into the comorbidity group; * *p* < 0.05, ** *p* < 0.001.

## Data Availability

The data that was support the finding of this study are available in National Health Insurance Research Database (NHIRD) in Taiwan. These data were available in the public domain for now.

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
