# Peer review of "Risk Role of Breast Cancer in Association with Human Papilloma Virus among Female Population in Taiwan: A Nationwide Population-Based Cohort Study"

_healthcare, 2022, doi:10.3390/healthcare10112235_

Round 1

Reviewer 1 Report

This is an important piece of research supporting the role of HPV in breast cancer. I have only one minor comment. In the introduction, the discussion on line 64-74 gets very technical for the non-medical/bench scientists and could use a bit of explanation. Some terms (e.g., BRCA) also could benefit form definition/explanation to reach a wider audience.

Author Response

Response to Reviewer 1 Comments

Point 1

In the introduction, the discussion on line 64-74 gets very technical for the non-medical/bench scientists and could use a bit of explanation. Some terms (e.g., BRCA) also could benefit form definition/explanation to reach a wider audience.

Response:

BRCA1 (BReast CAncer gene 1) and BRCA2 (BReast CAncer gene 2) that produce proteins that help repair damaged DNA. People who inherit harmful variants in one of these genes have increased risks of several cancers, especially breast cancer and ovarian cancer.

I will add some explanation to reach a wider audience. Appreciate your suggestion.

Reviewer 2 Report

Please continue your research so vital for society all over the world.

Readers will be delighted seeing your present and future publications.

Author Response

Point

Please continue your research so vital for society all over the world.

Readers will be delighted seeing your present and future publications.

Response

Thank you for your recognition and encouragement.

The etiology of breast cancer is still unclear, and we usually referred breast cancer as a multifactorial disease. We're trying to find out more ways to effectively prevent breast cancer. All authors will try to put it into practice in clinical work.

Reviewer 3 Report

Overall, the whole structure of this study is good and some corrections are recommended for providing clear information. Particularly, I listed the following comments in detail here.

In the abstract, the author needs to mention the ingredients of methods, and materials. Also, the finding of the assay could be added step by step based on material and method.

In introduction, the sentences are un-regularly disperse. Some sentences lack of references. For example, “Breast cancer is one of the cancers with the highest global prevalence among women and is the leading cause of death from cancer in women”, “However, the etiology of breast cancer remains unclear; breast cancer is generally referred to as a multifactorial disease.”, “Most of these factors are related to prolonged estrogen and progesterone stimulation”, “Human papillomavirus (HPV) infection is a sexually transmitted disease with the highest prevalence among both women and men.”, “In the HPV family, high-risk HPV types, including HPV16 and HPV18, have particularly been associated with an increased risk of cancer. The E6 and E7 proteins in HPV promote cellular proliferation and inactivate cell cycle checkpoint function, which then promotes viral proliferation by degrading p53 and Rb tumor suppressors. The E7 protein causes replication stress and oncogene-induced senescence.”, and so on.

Additionally, all of the names and terms should be completely mentioned for the first time, such as, BRCA.

Methods and Results: The authors need to mention the ingredients of methods, and materials.

Please add references to methods. Also, the finding of the assay could be added step by step based on material and method.

In discussion, discuss your own results before relating them to the results of other published work. Also, the authors must come their data step by step and comparison them with other study in Iran and other area.  Precise conclusion as it’s too short in its current form.

In the end, add a significant statement that must be structured as “what was offered by authors? Do the authors have more thoughts on this field?

Author Response

Point

In the abstract, the author needs to mention the ingredients of methods, and materials. Also, the finding of the assay could be added step by step based on material and method.

Response :

We analyzed data from the National Health Insurance Research Database (NHIRD) in Taiwan. Taiwan launched a single-payer National Health Insurance Program on March 1, 1995. As of 2007, 22.60 million of Taiwan’s 22.96 million population were enrolled in this program. NHIRD derived from this system by the Bureau of National Health Insurance, Taiwan and maintained by the National Health Research Institutes, Taiwan, are provided to scientists in Taiwan for research purposes.

Point

In introduction, the sentences are un-regularly disperse. Some sentences lack of references. For example, “Breast cancer is one of the cancers with the highest global prevalence among women and is the leading cause of death from cancer in women”, “However, the etiology of breast cancer remains unclear; breast cancer is generally referred to as a multifactorial disease.”, “Most of these factors are related to prolonged estrogen and progesterone stimulation”, “Human papillomavirus (HPV) infection is a sexually transmitted disease with the highest prevalence among both women and men.”, “In the HPV family, high-risk HPV types, including HPV16 and HPV18, have particularly been associated with an increased risk of cancer. The E6 and E7 proteins in HPV promote cellular proliferation and inactivate cell cycle checkpoint function, which then promotes viral proliferation by degrading p53 and Rb tumor suppressors. The E7 protein causes replication stress and oncogene-induced senescence.”, and so on.

Response  :

Breast cancer is one of the cancers with the highest global prevalence among women and is the leading cause of death from cancer in women

According to American cancer society Breast cancer is the most common cancer in women in the United States, except for skin cancers. It is about 30% (or 1 in 3) of all new female cancers each year

“However, the etiology of breast cancer remains unclear; breast cancer is generally referred to as a multifactorial disease.”, “Most of these factors are related to prolonged estrogen and progesterone stimulation”

According to WHO, half of breast cancers develop in women who have no identifiable breast cancer risk factor other than gender (female) and age (over 40 years).  Certain factors increase the risk of breast cancer including increasing age, obesity, harmful use of alcohol, family history of breast cancer, history of radiation exposure, reproductive history (such as age that menstrual periods began and age at first pregnancy), tobacco use and postmenopausal hormone therapy.

I had already mention some primary risk factor in the article.

Human papillomavirus (HPV) infection is a sexually transmitted disease with the highest prevalence among both women and men.

According to CDC (Central disease control and prevention) reveled HPV is the most common STI. There were about 43 million HPV infections in 2018, many among people in their late teens and early 20s. There are many different types of HPV. Some types can cause health problems, including genital warts and cancers.

Point

All the names and terms should be completely mentioned for the first time, such as, BRCA.

Response

BRCA1 (BReast CAncer gene 1) and BRCA2 (BReast CAncer gene 2) that produce proteins that help repair damaged DNA. People who inherit variants in genes have increased risks of several cancers, especially breast cancer and ovarian cancer.

I will add some explanation for the terms.

Point

In discussion, discuss your own results before relating them to the results of other published work. Also, the authors must come their data step by step and comparison them with other study in Iran and other area.  Precise conclusion as it’s too short in its current form.

Response :

Conclusion

The strengths of our study include a large sample size and robust event rate, ana-lytic control for known risk factors in the matched cohort, and independent confirma-tion of the magnitude and direction in the unmatched cohort.

Several HPV families, such as HPV16 and HPV18, have been linked to an in-creased risk of several types of cancer. In our study, we discovered these families to in-crease the risk of breast cancer.

People who were older than 49 years had a greater risk of developing breast can-cer. The risk of breast cancer for the 50−64 and the older than 65 age groups was re-spectively 1.67 and 1.36 times higher than that of people younger than 49 years. The patients with HPV had a higher risk of breast cancer than patients without HPV did, regardless of age group and the presence or absence of comorbidities. Moreover, the patients with HPV in the 50–64 age group were 1.39 times more likely to develop breast cancer than the patients without HPV in the same age group were. HPV may play a causal role in breast cancer.

According to the CDC reveled HPV is the most common sexually transmitted infec-tions. How to prevent HPV exposed is the way to prevent induce breast cancer risk. When used correctly and consistently, condoms offer one of the most effective methods of protection. HPV vaccines primarily benefit women who have not been exposed to HPV. Therefore, the vaccine’s actual effectiveness and protective effects against breast cancer remain unclear and warrant further investigation.

Point

In the end, add a significant statement that must be structured as “what was offered by authors? Do the authors have more thoughts on this field?

Response :

The prevalence of HPV in breast cancer indicates that prophylactic vaccination against HPV is required to limit infection in women. HPV vaccines primarily benefit women who have not been exposed to HPV. Therefore, the vaccine’s actual effectiveness and protective effects against breast cancer remain unclear and warrant further investigation.